# Junctional Epidermolysis Bullosa Caused by a Hemiallelic Nonsense Mutation in *LAMA3* Revealed by 18q11.2 Microdeletion

**DOI:** 10.3390/ijms26157343

**Published:** 2025-07-29

**Authors:** Matteo Iacoviello, Marilidia Piglionica, Ornella Tabaku, Antonella Garganese, Aurora De Marco, Fabio Cardinale, Domenico Bonamonte, Nicoletta Resta

**Affiliations:** 1Medical Genetics Unit, Department of Precision and Regenerative Medicine and Ionian Area (DiMePRe-J), University of Bari “Aldo Moro”, 70124 Bari, Italy; m.iacoviello3@gmail.com (M.I.); nelatabaku96@gmail.com (O.T.); 2Medical Genetics Unit, University Hospital Consortium Polyclinic of Bari, Piazza G. Cesare 11, 70124 Bari, Italy; marilidia.piglionica@policlinico.ba.it (M.P.); antonella.garganese@policlinico.ba.it (A.G.); 3Section of Dermatology and Venereology, Department of Precision and Regenerative Medicine and Ionian Area (DiMePRe-J), University of Bari “Aldo Moro”, 70124 Bari, Italy; aurorademarco94@gmail.com (A.D.M.); domenico.bonamonte@uniba.it (D.B.); 4Department of Pediatrics, Giovanni XXIII Pediatric Hospital, Via G. Amendola 207, 70126 Bari, Italy; fabiocardinale1961@gmail.com; 5Section of Dermatology and Venereology, University Hospital Consortium Polyclinic of Bari, Piazza G. Cesare 11, 70124 Bari, Italy

**Keywords:** junctional epidermolysis bullosa, laryngo-onycho-cutaneous syndrome, *LAMA3*

## Abstract

Inherited epidermolysis bullosa (EB) is a heterogeneous clinical entity that includes over 30 phenotypically and/or genotypically distinct inherited disorders, characterized by mechanical skin fragility and bullae formation. Junctional EB (JEB) is an autosomal recessive disease characterized by an intermediated cleavage level within the skin layers, commonly at the “lamina lucida”. Laryngo-onycho-cutaneous syndrome (LOC) is an extremely rare variant of JEB, characterized by granulation tissue formation in specific body sites (skin, larynx, and nails). Although most cases of JEB are caused by pathogenic variants occurring in the genes encoding for classical components of the lamina lucida, such as laminin 332 (*LAMA3*, *LAMB3*, *LAMC2*), integrin α6β4 (*ITGA6*, *ITGB4*), and collagen XVII (*COL17A1*), other variants have also been described. We report the case of a 4-month-old male infant who presented with recurrent bullous and erosive lesions from the first month of life. At the first dermatological evaluation, the patient was agitated and exhibited hoarse breathing, a clinical sign suggestive of laryngeal involvement. Multiple polygonal skin erosions were observed on the cheeks, along with similar isolated, roundish lesions on the scalp and legs. Notably, nail dystrophy and near-complete anonychia were evident on the left first and fifth toes. Due to the coexistence of skin erosions and nail dystrophy in such a young infant, a congenital bullous disorder was suspected, prompting molecular analysis of all potentially involved genes. In the patient’s DNA, clinical exome sequencing (CES) identified a pathogenic variant, apparently in homozygosity, in the exon 1 of the *LAMA3* gene (18q11.2; NM_000227.6): c.47G > A;p.Trp16*. The presence of this variant was confirmed, in heterozygosity, in the genomic DNA of the patient’s mother, while it was absent in the father’s DNA. Subsequently, trio-based SNP array analysis was performed, revealing a paternally derived pathogenic microdeletion encompassing the *LAMA3* locus (18q11.2). To our knowledge, this is the first reported case of JEB with a LOC-like phenotype caused by a maternally inherited monoallelic nonsense mutation in *LAMA3*, unmasked by an almost complete deletion of the paternal allele. The combined use of exome sequencing and SNP array is proving essential for elucidating autosomal recessive diseases with a discordant segregation. This is pivotal for providing accurate genetic counseling to parents regarding future pregnancies.

## 1. Introduction

Inherited epidermolysis bullosa (EB) is a variegated clinical entity comprising over 30 phenotypically and/or genotypically distinct diseases, all characterized by a shared tendency toward mechanical skin fragility and blister formation [1,2]. Depending on skin cleavage level, three major EB variants have been described: EB simplex (EBS), junctional EB (JEB), and dystrophic EB (DEB) [1,3]. Due to chronic and recurrent wounds with impaired tissue regeneration, EB patients are generally at higher risk of developing cutaneous squamous cell carcinoma (cSCC) compared to the general population [1]. The risk for cSCC in EB appears to correlate with disease severity: it is rare in EBS patients (2.6%), but more frequent in JEB and DEB [1]. In JEB, the incidence of cSCC ranges from 6.8% to 16.2% [1]. Beyond its epidemiological relevance, cSCC in EB patients is also a critical determinant of survival, as it tends to exhibit highly aggressive behavior, often representing the leading cause of death [1,4]. Although each form of EB may significantly impair quality of life and exhibit various degrees of severity [5], JEB is considered the most critical, particularly its “severe” subtype, which is associated with early lethality in the first 6 to 24 months of life [3].

JEB is characterized by an intermediate level of cleavage within the skin layers, typically at the “lamina lucida” [3]. Key clinical features of JEB include the development of exuberant granulation tissue in perioral, axillary, and neck regions, as well as dental enamel hypoplasia and nail dystrophy [2]. This form of EB is classified into two main subtypes, each with distinct clinical courses [2,3]. Severe JEB is characterized by extensive mucocutaneous blistering and extracutaneous complications, often leading to early mortality within the first few years of life [6]. In contrast, intermediate JEB presents with a more variable clinical course, with affected individuals typically surviving into adulthood [3]. This form shows considerable heterogeneity in severity and tissue involvement [3]. Among the rarest variants of JEB, laryngo-onycho-cutaneous syndrome (LOC) is defined by granulation tissue formation at specific anatomical sites, particularly the skin, larynx, and nails [7]. Clinically, skin blistering with aberrant granulation tissue on chronic erosions, especially on the cheeks, ears, neck, and axillary, is highly suggestive [7]. Granulation tissue may also affect distal digits, leading to periungual inflammation, nail dystrophy, and anonychia. Periocular involvement may also occur, often with progressive latero-medial extension and risk of symblepharon [7]. Crucially, laryngeal involvement with subglottic stenosis due to excessive granulation is a hallmark of LOC, potentially resulting in life-threatening respiratory complications [7].

From a molecular perspective, JEB is a genetically heterogeneous disease caused by biallelic mutations in genes encoding structural proteins essential to the integrity of the basement membrane zone [2]. These include type XVII collagen (*COL17A1*), laminin 332 (*LAMA3*, *LAMB3*, *LAMC2*), integrin α6β4 (*ITGA6*, *ITGB4*), and the integrin α3 subunit (*ITGA3*) [2,3]. Among these, laminin 332 is the most frequently disrupted [3]. This heterotrimeric protein, secreted by basal keratinocytes into the extracellular space, plays a pivotal role in forming anchoring filaments within the lamina lucida [2]. As a key component of the epidermal basement membrane, it contributes to keratinocyte adhesion and structural integrity [2]. Laminin 332 is found not only in the epidermal basement membrane but also in various epithelial basement membranes beyond the skin, including those of the cornea, kidney, lung, thymus, brain, and gastrointestinal tract [8]. This widespread distribution explains the extensive mucocutaneous adhesion and extracutaneous involvement observed in laminin 332-deficient JEB [2]. In addition to its structural role, laminin 332 regulates keratinocyte migration, playing a central role in wound healing [8].

The *LAMA3* gene, which encodes one of the subunits of laminin 332, is located on chromosome 18q11.2, and generates three major alternative transcripts encoding the laminin α3 isoforms: α3a, α3b1, and α3b2 [9]. Among these, LAMA3A and LAMA3B are the main transcripts encoding the α3a and α3b1 isoforms, respectively. LAMA3A is transcribed from an internal promoter located within intron 38, with protein coding from exons 39 to 76, producing a 5175 bp open reading frame (ORF) and a 1724-amino-acid protein. Conversely, LAMA3B includes exons 1 to 38 and the shared exons 40 to 76 (skipping exon 39), resulting in a 10,002 bp ORF and 3333-amino-acid proteins (Figure 1B) [9]. In addition, at the mRNA level, about 20% of the LAMA3B transcript also skips exon 10: the full-length transcript of the LAMA3B isoform has been termed LAMA3B1, encoding laminin α3B1, and the shorter transcript of the LAMA3B isoform has been termed LAMA3B2, encoding laminin α3B2. Most mutations found in patients with severe JEB result in premature termination codons (PTCs) in both LAMA3A and LAMA3B transcripts, suggesting that the loss of both α3a and α3b is responsible for the severe phenotype [9]. In contrast, selective loss of α3a alone has been proposed as the molecular basis of LOC [10].

Although all JEB subtypes, including LOC, are autosomal recessive (AR) disorders caused by biallelic pathogenic variants in a single gene, AR diseases can also manifest in individuals with a single detectable pathogenic allele if the second allele is disrupted, e.g., by deletion, uniparental disomy, or deep intronic variants. In this context, we present the clinical and molecular findings for a patient affected by JEB with a LOC-like phenotype, resulting from a hemiallelic nonsense mutation in *LAMA3* in conjunction with a chromosome 18 microdeletion. To the best of our knowledge, this is the first reported case of JEB arising from this combination of genetic mechanisms.

## 2. Results

### 2.1. Clinical Features and Findings

A four-month-old male infant presented with a history of chronic recurrent skin erosions and blistering primarily affecting the face and lower limbs. At the initial dermatological evaluation, multiple polygonal erosions were observed on the cheeks, with isolated, rounder lesions on the lower limbs and scalp (Figure 2A–C). Nail dystrophy and near-complete anonychia were noted on the first and fifth toes of the left foot, along with hoarse respiration (Figure 2D). Due to the roundish shape of the erosions located on the scalp and limbs, a bullous origin was hypothesized, prompting consideration of a neonatal bullous disorder. A skin swab from the facial lesions, along with laboratory and serological tests, excluded a bacterial bullous infection. Furthermore, autoantibodies associated with common autoimmune bullous diseases (including autoantibodies against bullous pemphigoid antigens 180 and 230, autoantibodies against desmoglein 1 and 3, and autoantibodies against type VII collagen) were within normal limits, thereby increasing the likelihood of a congenital rather than an acquired etiology. Neither parent presented clinical signs compatible with cutaneous diseases. Moreover, due to an increase in breathing hoarseness, an ENT consultation (Ear, Nose, and Throat) was carried out, excluding life-threatening laryngeal obstructions and describing mild mucosal erosions.

### 2.2. Molecular Analysis

Clinical exome sequencing (CES) revealed an apparently homozygous pathogenic variant, affecting the *LAMA3* gene (18q11.2; OMIM *600805). The variant was predicted to exert a different effect on LAMA3A and LAMA3B1, the two main *LAMA3* transcript isoforms. It was located in exon 1 of the LAMA3A transcript isoform (NM_000227.6; c.47G > A;p.Trp16*—Figure 3A), leading to a premature stop codon, as well as in intron 38 of the LAMA3B1 isoform (NM_198129.4; c.4998 + 1430G > A;p.?), a deep intronic position. Parental testing via direct sequencing showed the mother was heterozygous for the variant, while the father was a non-carrier (Figure 3B). Trio-based SNP array analysis was subsequently performed, revealing a paternally inherited microdeletion involving the 18q11.2 region, arr[GRCh37] 18q11.2(21,394,409_21,716,773)x1pat, a deletion of approximately 322 kb (Figure 1A). This pathogenic deletion encompasses three genes: *LAMA3*, *TTC39C*, and *RNU5A-6P*. In particular, the microdeletion corresponds to the specific promoter and the whole coding sequence of the LAMA3A isoform and to the 15–75 exons of LAMA3B isoforms.

## 3. Discussion

We describe a case of JEB with a LOC-like phenotype, resulting from a hemiallelic nonsense mutation in *LAMA3* in combination with a pathogenic microdeletion on chromosome 18. The most striking clinical features included polygonal erosions in the head and neck area, and progressive nail dystrophy due to granulation tissue over the nail beds (Figure 2A–C). The combination of skin erosions and nail abnormalities in a neonate raised suspicion of inherited EB. In addition, hoarse breathing, a key clinical indicator of laryngeal involvement, strongly supported a diagnosis consistent with the LOC phenotype.

Although diagnostic techniques such as immunofluorescence antigen mapping and electron microscopy can expedite diagnosis, skin biopsies in EB patients are often challenging and potentially harmful due to impaired wound healing. For this reason, molecular testing was prioritized in our case, also to avoid general anesthesia and potential airway trauma from intubation. We identified an individual with a compound heterozygous genotype for two pathogenic, presumably loss-of-function, variants. Pathogenic or likely pathogenic variants, whether in a homozygous or compound heterozygous state, are associated with both severe (OMIM #619784) and mild (OMIM #619783) forms of JEB, as well as with LOC (OMIM #245660) [3,7]. To date, more than 250 pathogenic or likely pathogenic variants have been reported in *LAMA3* (https://varsome.com/gene/hg38/LAMA3, accessed on 12 June 2025) (Figure 4). c.47G > A;p.Trp16* has also been reported in a homozygous state in individuals with JEB [9].

Given the parental genotypes, the heterozygous mother and non-carrier father, offspring would be expected to be either non-carriers or heterozygous carriers. However, our patient was found to be apparently homozygous. Assuming confirmed paternity, this paradox suggests two possibilities: either a deletion involving the paternal *LAMA3* allele, or maternal uniparental disomy. SNP array analysis confirmed the first hypothesis, revealing a paternally inherited microdeletion encompassing *LAMA3*. Consequently, the patient carries compound heterozygous pathogenic variants: a maternally inherited nonsense mutation and a paternally inherited microdeletion. This finding is consistent with the LOC-like phenotype observed in our patient. Indeed, the hemiallelic nonsense variant specifically affects the LAMA3A transcript, while the microdeletion likely disrupts the expression of both principal *LAMA3* transcripts—an assumption further supported by the fact that the deletion also encompasses several *LAMA3* enhancer sequences (https://genome.ucsc.edu/index.html accessed on 23 July 2025). The other two deleted genes (*TTC39C* and *RNU5A-6P*) are not currently associated with any disease phenotypes and are unlikely to contribute to the clinical presentation.

To date, therapy for JEB is primarily supportive [11]. Dry extracts from Betula pendula Roth and Betula pubescens are currently the only approved topical treatments for patients aged 6 months and older with DEB and JEB (https://www.ema.europa.eu/en/medicines/human/EPAR/filsuvez accessed on 23 July 2025). Meanwhile, preclinical and clinical trials are ongoing to develop targeted therapeutic strategies. Approaches under investigation include readthrough of premature termination codons (PTCs) [11,12,13,14,15], RNA-based therapies (e.g., exon skipping with antisense oligonucleotides [16]), lentiviral gene transfer, and protein misfolding [17]. For these therapies to be successful, accurate molecular diagnosis and mutation stratification are essential. Thus, genotype-based subclassification and precision medicine approaches represent a critical frontier in EB research [18]. In our patient, who was under 6 months of age at the time of first evaluation, only mild topical treatment was initiated. Management includes eosin-containing skin solutions to reduce granulation and absorbent foam dressing to protect the skin and accelerate wound healing. Clinical monitoring remains essential to track wound healing, oxygen saturation, and early signs of systemic or cutaneous complications. Indeed, the patient has experienced worsening of granulation tissue on facial erosions and new nail involvement affecting the contralateral toes and both hands (Figure 2E–I). Although hoarse breathing has persisted, no life-threatening airway obstruction has occurred thus far, although the present case suggests the importance of a multidisciplinary approach in diagnosing and managing complex and rare cases, especially when syndromic with multi-organ involvement. To the best of our knowledge, this is the first reported case of JEB with a LOC-like phenotype caused by a hemiallelic nonsense mutation in *LAMA3*, unmasked by a pathogenic microdeletion. Genetic analysis was crucial to establish the diagnosis. Moreover, this case highlights the essential role of combining exome sequencing with copy number analysis (SNP array) in resolving cases of autosomal recessive inheritance with discordant segregation patterns. This approach is crucial for providing accurate genetic counseling. Since the microdeletion was inherited (not de novo) from the father, the recurrence risk for JEB in future pregnancies is 25%, consistent with a classical autosomal recessive inheritance model.

## 4. Materials and Methods

### 4.1. Patient and Family Members

This is a retrospective description of the clinical and molecular profile of an infant presenting with JEB with a LOC-like phenotype and his family. The parents gave their informed consent. This work was conducted in accordance with the principles of the Helsinki Declaration. The patient was hospitalized due to their clinical condition and underwent a comprehensive dermatological evaluation. Laboratory and serological tests were performed to investigate the presence of potential bacterial skin infections or circulating autoantibodies specific to the most common autoimmune bullous diseases. Furthermore, clinical exome sequencing (CES) was performed on the patient’s DNA while the SNP array analysis was conducted on the “trios”.

### 4.2. Genomic DNA (gDNA) Sample Extraction

gDNA was extracted from the proband’s and their parents’ peripheral blood (PB) samples by using a QIAamp Mini Kit (Qiagen, Hilden, Germany) and gDNA was quantified with a BioSpectrometer Plus instrument (Eppendorf, Hamburg, Germany), following the manufacturer’s instructions.

### 4.3. Clinical Exome Sequencing (CES)

CES on the proband’s DNA was performed using the TruSight One Expanded Sequencing Panel kit (Illumina, San Diego, CA, USA). gDNA concentration and quality were evaluated by using a Qubit dsDNA HS Assay Kit on a Qubit 2.0 Fluorimeter (Invitrogen, Carlsbad, CA, USA), following the manufacturer’s instructions. Libraries were prepared by using the NextEra Flex for Enrichment protocol (Illumina) and were sequenced on the NextSeq550Dx Illumina platform (Illumina). Data analysis was performed by using the NextSeq control software v.4.2.0 and Local Run Manager software v4.0.0, both provided by Illumina (Illumina). Reads were aligned against the human genome reference (GRCh37/hg19) by the Burrows-Wheeler Aligner (BWA) Aligner software v.11.5 [19]. Variant calling was performed using the Genome Analysis Toolkit (GATK) [20]. A mean coverage depth of 154X was obtained. Variant calling data were analyzed with Geneyx analysis software v. 6.1.1 (Geneyx, Herzliya, Israel). Variants were filtered and prioritized by utilizing the HPO term [21] “Epidermolysis bullosa”, using in silico tools (Alamut Visual Plus, Combined Annotation Dependent Depletion (CADD) v.1.6) [22] and public databases (ClinVar, https://www.ncbi.nlm.nih.gov/clinvar accessed on 12 June 2025; LOVD, https://www.lovd.nl/; Varsome, https://varsome.com/; Franklin by Genoox, https://franklin.genoox.com/clinical-db/home accessed on 12 June 2025; OMIM, https://www.omim.org/), and were classified following the ACMG criteria [23]. Only genes related to clinical indication were investigated. Binary Alignment Map (BAM) files were visually inspected by the Integrative Genome Viewer software 2.19.4 (IGV) and Alamut Visual Plus Genome Viewer (Sophia Genetics, Lausanne, Switzerland), and variants were reported according to the Human Genome Variants Society (HGVS) recommendations [24].

### 4.4. Sanger Sequencing Analysis

The variant identified by CES was confirmed by Sanger sequencing (Applied Biosystem, Waltham, MA, USA) on SeqStudio Genetic Analyzer on the proband’s and their parents’ DNA, according to the manufacturer’s instructions.

### 4.5. SNP Array Analysis

A trio-based SNP array (Illumina Infinium^tm^ CytoSNP 850K v1.4) was performed, using the Infinium CytoSNP-850K BeadChip, which contains 848,902 selected single-nucleotide polymorphisms (SNPs) spanning the genome with enriched coverage for 3262 genes of known relevance (Illumina, Inc.), following the manufacturer’s instructions. The data were analyzed using BlueFuseTM Multi v4.5 software (BlueGnome^®^, BlueGnome Ltd. (now part of Illumina, Inc.), Cambridge, UK). The genomic positions are presented as mapped to the human reference genome build (GRCh37/hg19). The results were reported according to the American College of Medical Genetics and Genomics (ACMG) guidelines [25], classified as benign, likely benign, variant of uncertain significance (VUS), likely pathogenic, or pathogenic, and the following databases of copy number variation (CNV): the Database of Genomic Variants (DGV, http://projects.tcag.ca/variation/ accessed on 12 June 2025), the Database of Chromosomal Imbalance and Phenotype in Humans Using Ensembl Resources (DECIPHER, http://decipher.sanger.ac.uk/), the Clinical Genome Resource (ClinGen, https://clinicalgenome.org/), and the UCSC Genome Bioinformatics database (http://genome.ucsc.edu). This allowed the accurate profiling of chromosomal aberrations, such as duplications, deletions, unbalanced rearrangements, and copy-neutral absence of heterozygosity (AOH) events. Variants were annotated according to ISCN 2024 nomenclature [26].

## Figures and Tables

**Figure 1 ijms-26-07343-f001:**
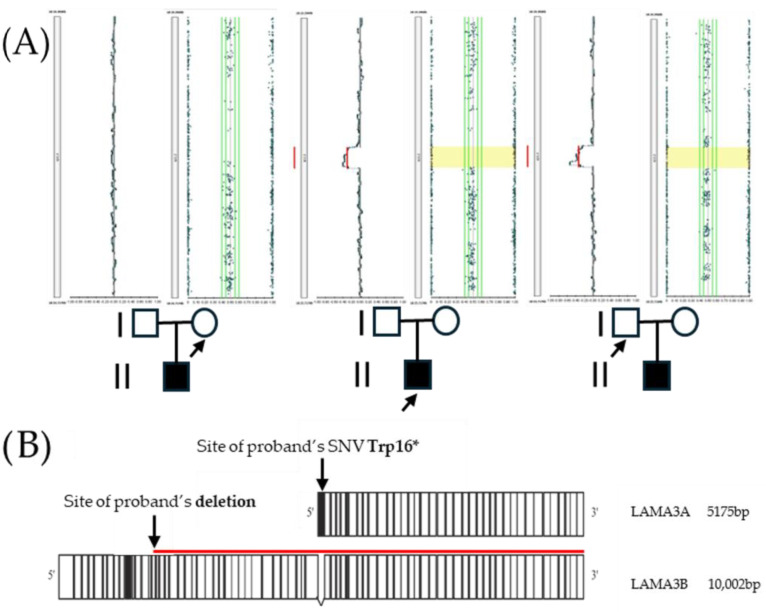
Trio-based SNP array analysis detected a paternally inherited pathogenic microdeletion of 322 kb involving the *LAMA3* locus (18q11.2) in the proband’s DNA. (**A**) Pedigree of studied individuals: male, female, and affected proband are represented by square, circle, and shading, respectively; arrows indicate the individuals who underwent genetic testing. (**B**) The two main transcripts of the *LAMA3* gene, encoding the α3a and α3b1 isoforms, were both affected: the maternally inherited nonsense mutation on LAMA3A isoform and the paternally inherited microdeletion on LAMA3B isoforms.

**Figure 2 ijms-26-07343-f002:**
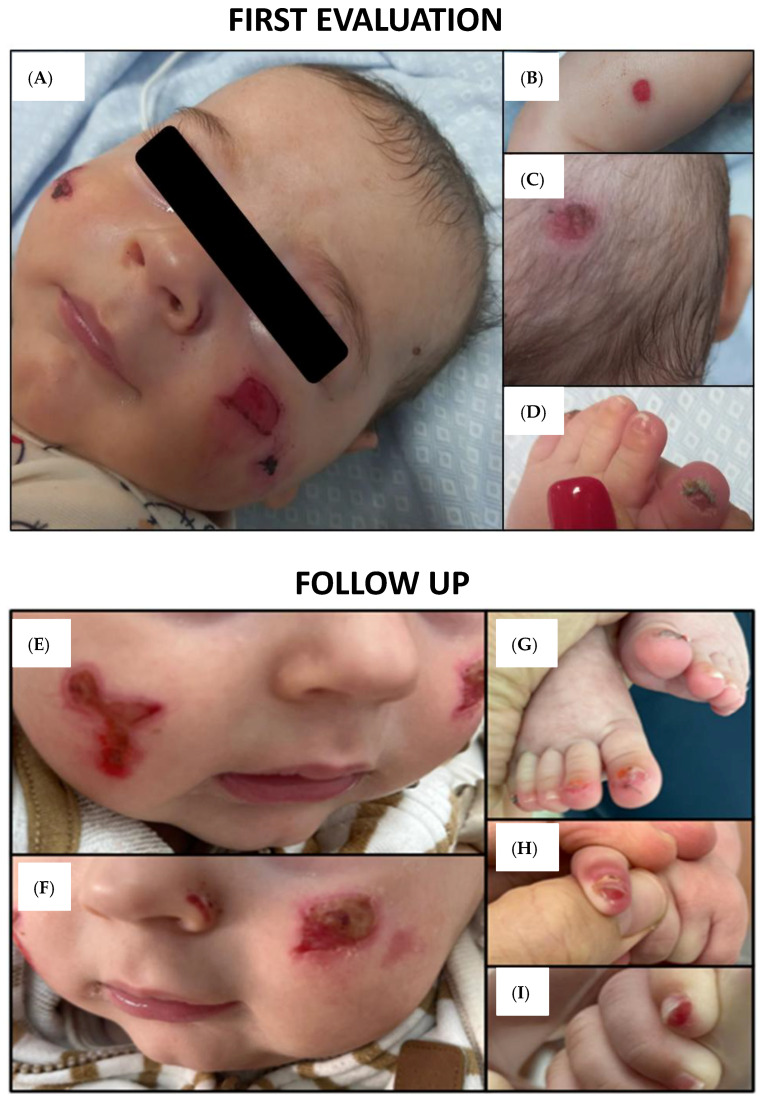
Clinical evolution over time: the “First Evaluation” panel (upper) illustrates the initial clinical presentation of our 4-month-old patient, while the “Follow-Up” panel (lower) shows the evolution of the lesions over time. (**A**) Polygonal granulating erosions on the cheek; (**B**,**C**) further isolated roundish skin erosions on the patient’s right lower limb and scalp, respectively; (**D**) nail dystrophy with almost complete anonychia involving the patient’s first and fifth left toes; (**E**,**F**) worsening of granulation on previous erosions on the cheeks; (**G**) progression of nail dystrophy involving toes with diffuse almost complete anonychia; (**H**,**I**) development of subungual granulation tissue causing progressive nail dystrophy on both hands.

**Figure 3 ijms-26-07343-f003:**
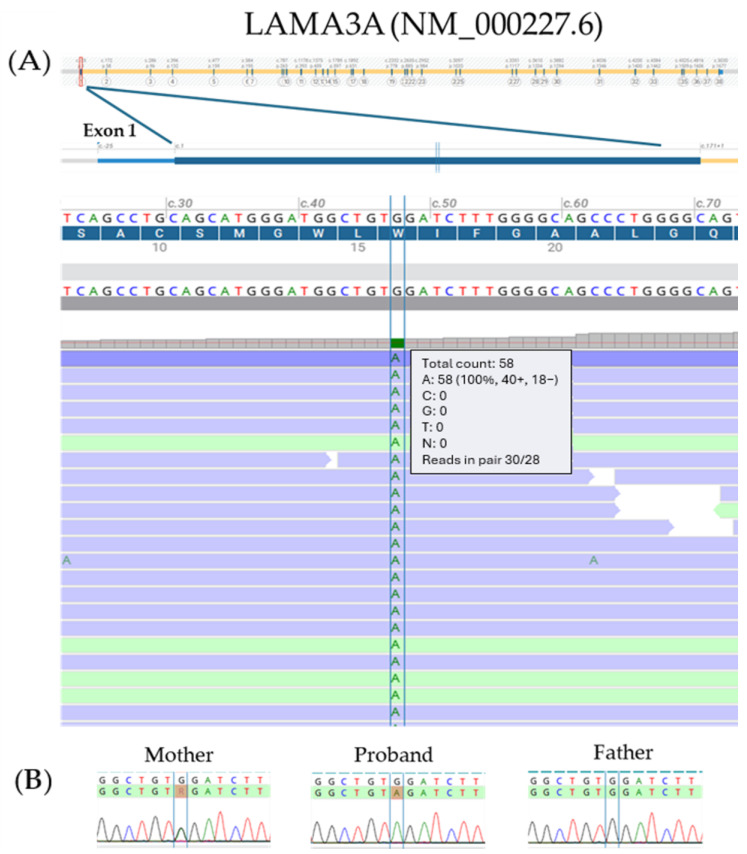
The molecular analysis of the proband’s DNA revealed an apparently homozygous pathogenic variant, c.47G > A;p.Trp16*, located in exon 1 of the *LAMA3* gene (NM_000227.6). (**A**) Exon–intron organization of the LAMA3A isoform and read alignment in the proband’s DNA from the clinical exome sequencing (CES) investigation. (**B**) Parental testing via Sanger sequencing showed that the mother was heterozygous for the same variant, while the father was a non-carrier (A: adenine; T: thymine; C: cytosine; G: guanine; R: adenine or guanine).

**Figure 4 ijms-26-07343-f004:**
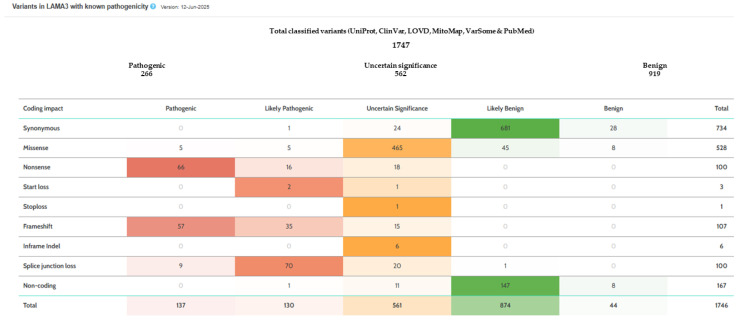
Genetic variant distribution in *LAMA3* (*NM_198129.4*). Adapted from VarSome (https://varsome.com/gene/hg38/LAMA3 accessed on 12 June 2025).

## Data Availability

The original contributions presented in this study are included in the article. Further inquiries can be directed to the corresponding author.

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
