# Peer review of "Junctional Epidermolysis Bullosa Caused by a Hemiallelic Nonsense Mutation in LAMA3 Revealed by 18q11.2 Microdeletion"

_ijms, 2025, doi:10.3390/ijms26157343_

Round 1
Reviewer 1 Report
Comments and Suggestions for Authors
Iacoviello et al. report a case with junctional epidermolysis bullosa with the rare LOC subtype deciphering an unusal genetical setting. This is a very interesting case showing the importance of a sufficient workup to establish the diagnosis. However, a few points could be improved:
- The authors described ruling out a superinfection and autoimmune bullous diseases stating “autoantibodies associated with common autoimmune bullous diseases were within normal limits“. The individual autoantigens investigated should be briefly mentioned.
- Figure 4, please explain “R” in contrast to ACGT in the figure legend for readers not familiar with genetics.
- The position of Figure 2 seems odd, as subsequent analysis to the sanger sequencing approach it would be placed better after Fig 4.
- The authors should make a short comment on ENT consultation given the hoarseness and highlight the importance of interdisciplinary care for these fragile patient group.
Author Response
1) Comment: “The authors described ruling out a superinfection and autoimmune bullous diseases stating ‘autoantibodies associated with common autoimmune bullous diseases were within normal limits’. The individual autoantigens investigated should be briefly mentioned.”
Response:
We agree with the reviewer’s suggestion and have now specified the tested circulating autoantibodies in the manuscript. This clarification has been added in section 3.1. Clinical features and findings (lines 205–207).
2) Comment: “Figure 4, please explain ‘R’ in contrast to ACGT in the figure legend for readers not familiar with genetics.”
Response:
We thank the reviewer for pointing this out. The figure legend (now corresponding to Figure 3 following the rearrangement described in point 3) has been expanded to define the nucleotide abbreviations used. The legend now includes: “A: adenine; T: thymine; C: cytosine; G: guanine; R: adenine or guanine.”
3) Comment: “The position of Figure 2 seems odd, as subsequent analysis to the Sanger sequencing approach it would be placed better after Fig 4.”
Response:
The figure mentioned in the comment does not show primary data from our study but provides an overview of known LAMA3 variants. In light of the reviewer’s comment, we have relocated this figure to follow Figure 4, but also decided to place it after the Discussion section to better reflect its contextual purpose. The in-text citation has been moved accordingly (now in lines 301–303). To maintain consistency across the manuscript, all figures have been renumbered following this reorganization: the original Figures 2 and 4 are now Figures 4 and 3, respectively.
4) Comment: “The authors should make a short comment on ENT consultation given the hoarseness and highlight the importance of interdisciplinary care for these fragile patient group.”
Response:
We appreciate this important suggestion. A brief note about the ENT evaluation has been added in section 3.1. Clinical features and findings (lines 209–211). Furthermore, we have included a statement on the significance of interdisciplinary management in the Discussion section (lines 339–341).
Reviewer 2 Report
Comments and Suggestions for Authors
This study report a case of JEB caused by compound heterozygous mutations in the LAMA3 gene (a maternally inherited nonsense mutation + a paternally inherited microdeletion). This case provides clear molecular diagnostic value and genetic insights, offering new perspectives on the genotype-phenotype correlations and diagnostic strategies for JEB.
- It is necessary to clarify whether the microdeletion region encompasses regulatory elements of LAMA3(e.g., promoter/enhancer) to substantiate its comprehensive disruptive effect on gene expression (currently, only "containing the promoter and exons" is mentioned).
- It should be explicitly stated whether PCR amplification or MLPA validation of the microdeletion breakpoints was performed to exclude the possibility of SNP array false positives.
- Figures 3 and 5 all show nail dystrophy/anonychia. It is recommended to integrate these into chronological comparative panels (e.g., "Initial Presentation vs. Follow-up") to highlight disease progression.
- The entire text needs to uniformly use HGVS nomenclature: For example, "c.[47G>A]" should be changed to "c.47G>A" (removing the square brackets, as they are not required for phase-unknown compound heterozygosity)
Author Response
1) Comment: “It is necessary to clarify whether the microdeletion region encompasses regulatory elements of LAMA3 (e.g., promoter/enhancer) to substantiate its comprehensive disruptive effect on gene expression (currently, only ‘containing the promoter and exons’ is mentioned).”
Response:
Following consultation of the UCSC Genome Browser, we have confirmed that the microdeletion includes several enhancer regions of LAMA3, in addition to the promoter, the full coding sequence of the LAMA3A isoform, and exons 15 to 75 of the LAMA3B isoforms. We have accordingly revised the relevant sentence in the Discussion (lines 313–317).
2) Comment: “It should be explicitly stated whether PCR amplification or MLPA validation of the microdeletion breakpoints was performed to exclude the possibility of SNP array false positives.”
Response:
We appreciate the reviewer’s concern regarding the need to confirm the microdeletion. Although PCR or MLPA validation was not performed in our laboratory, we consider a false positive highly unlikely. The microdeletion was independently identified by SNP-array analysis in both the patient and his father, as part of a trio-based diagnostic approach, thereby confirming its segregation. Furthermore, the same microdeletion was independently detected in the patient’s paternal uncle (i.e., the father’s brother) by a separate diagnostic center using SNP-array. This convergence of results across three related individuals and two independent laboratories provides robust confirmation of the microdeletion’s presence.
3) Comment: “Figures 3 and 5 all show nail dystrophy/anonychia. It is recommended to integrate these into chronological comparative panels (e.g., ‘Initial Presentation vs. Follow-up’) to highlight disease progression.”
Response:
We agree that a comparative layout improves clarity. As recommended, we have merged Figures 3 and 5 into a unified composite figure (now labeled Figure 2). which directly follows the section 3.1. Clinical features and findings. Image labels from the former Figure 5 have been adjusted to continue the alphabetical sequence of the previous figure. All figure citations in the manuscript have been updated accordingly.
4)Comment: “The entire text needs to uniformly use HGVS nomenclature: For example, "c.[47G>A]" should be changed to "c.47G>A" (removing the square brackets, as they are not required for phase-unknown compound heterozygosity).”
Response:
We thank the reviewer for pointing this out. The text has been revised to ensure consistency with HGVS nomenclature, and square brackets have been removed accordingly in line 39.